# Unraveling the Association: How Identity Mediates the Impact of Childhood Trauma on Criminal Behavior

**DOI:** 10.3390/bs15010056

**Published:** 2025-01-09

**Authors:** Stefan Bogaerts, Deni Tressová, Esmée Feijen, Marija Janković

**Affiliations:** 1Department of Developmental Psychology, Tilburg University, 5037 AB Tilburg, The Netherlands; s.bogaerts@tilburguniversity.edu (S.B.); d.tressova@tilburguniversity.edu (D.T.); esmeefeijen2000@gmail.com (E.F.); 2Fivoor Science and Treatment Innovation (FARID), 3014 AE Rotterdam, The Netherlands

**Keywords:** identity, childhood trauma, criminal behavior, consolidated identity, disturbed identity, lack of identity

## Abstract

Previous research has consistently linked childhood trauma to criminal behavior in adulthood, yet the mechanisms driving this association remain poorly understood. This study investigated whether identity mediates this relationship, focusing on three identity constructs: consolidated identity, disturbed identity, and lack of identity. Criminal behavior was operationalized as a dichotomous variable, distinguishing between 103 community participants (53.9%) and 88 forensic psychiatric patients (46.1%) in a sample of 191 male participants (*M*age = 39.82, *SD*age = 14.14). Mediation analysis was conducted using PROCESS macro model 4, controlling for age and overall personality dysfunction. The results demonstrated that childhood trauma was associated with adult criminal behavior. Additionally, childhood trauma was positively associated with lack of identity but showed no significant effect on consolidated and disturbed identity. Likewise, lack of identity was the only identity variable associated with criminal behavior and emerged as the sole mediator between childhood trauma and criminal behavior. These findings underscore the important role of identity, particularly the lack of identity, in understanding pathways to criminal behavior. Interventions aimed at strengthening individuals’ sense of self may help mitigate criminal tendencies in individuals with a history of childhood trauma, though longitudinal research is needed to further validate these findings.

## 1. Introduction

Childhood trauma is a pervasive societal issue with profound and long-lasting negative consequences on individuals’ development and well-being. Globally, its prevalence is estimated at 17.0% ([64]), while, in the Netherlands, around 3% of all children and adolescents experience traumatic events, a figure likely underestimated due to underreporting ([47]). Childhood trauma encompasses exposure to harmful events before the age of 18, including emotional, physical, or sexual abuse, as well as emotional and physical neglect ([6]). Research has consistently linked childhood trauma to adverse outcomes in later life, such as substance abuse, low self-esteem, aggression, and behavioral problems ([23]; [28]; [48]; [55]). The impact is more pronounced when multiple forms of childhood trauma are experienced ([16]; [20]). This study specifically focuses on the cumulative effects of the five types of childhood trauma mentioned.

### 1.1. Childhood Trauma and Criminal Behavior

Childhood trauma has been strongly associated with antisocial and criminal behavior ([40]; [51]; [52]). Numerous studies have reported a high prevalence of childhood trauma in individuals within the criminal justice system. For instance, [44] ([44]) found that 75% of high-security forensic inpatients had experienced childhood trauma, with 65% exposed to multiple types. Additionally, childhood trauma was also a significant predictor of life course persistent offending, with individuals who experienced trauma more likely to engage in chronic criminal behavior ([18]; [29]). Several theoretical frameworks have attempted to explain the link between childhood trauma and criminal behavior. From an attachment theory perspective, early traumatic experiences can lead to insecure attachments, impairing the ability to form healthy relationships and regulate emotions ([27]; [36]; [30]). This can result in difficulties in emotion regulation and cognitive distortions, such as hostile attribution bias, which, in turn, may contribute to criminal behavior ([14]). Social learning theory provides another lens, suggesting that violence and aggression are transmitted intergenerationally through observation and imitation of violent role models ([59]). While these frameworks are supported by empirical evidence ([8]; [35]), the specific mechanisms underlying this relationship remain poorly understood ([37]). Particularly, there is a limited understanding of whether identity development explains the link between childhood trauma and criminal behavior.

### 1.2. Identity Development

[25] ([25]) defined identity as the pursuit of uniqueness and consistency across roles and contexts. Identity formation, a key developmental task during adolescence and early adulthood ([24]), spans from identity confusion, characterized by a lack of purpose and difficulty with life commitments, to identity synthesis, where consistent beliefs and values establish a stable sense of self ([10]; [26]; [38]). Building on Erikson’s work, [42] ([42]) identified two core dimensions of identity: exploration and commitment, later expanded by [38] ([38], [39]) to include five dimensions: commitment making, identification with commitment, exploration in breadth, exploration in depth, and ruminative exploration. In addition to these dimensional conceptualizations, identity can also be viewed from a clinical perspective ([33]). Since part of our sample includes forensic psychiatric patients, we have chosen to adopt this clinical conceptualization of identity. In our study, identity was operationalized through three variables: consolidated identity, reflecting identity synthesis, and two aspects of identity confusion: disturbed identity and lack of identity ([33]). A consolidated identity reflects a well-developed, coherent, and stable sense of self, where values and experiences are aligned. In contrast, a disturbed identity involves difficulties integrating various aspects of self-concept, accompanied by uncertainty about values, opinions, and beliefs. This can lead to confusion, self-doubt, and an excessive reliance on others for validation. Finally, a lack of identity signifies a complete disconnection from one’s sense of self, resulting in feelings of being lost, undefined, and empty. In summary, these levels reflect stages of personal identity development, ranging from the stability and clarity of consolidated identity to the instability and vagueness of a lack of identity ([33]).

### 1.3. Childhood Trauma and Identity Formation

Childhood trauma can disrupt identity development by impairing basic trust and self-concept, which often results in identity confusion ([5]; [34]; [53]). Emotional abuse and neglect, in particular, negatively impact emotion regulation, a critical factor in forming a coherent identity ([21]; [32]). Moreover, the absence of stable role models during key developmental stages, frequently a consequence of childhood trauma, severely limits opportunities for establishing a coherent sense of self ([3]; [26]; [50]). Together, these findings highlight how childhood trauma can undermine identity formation, often leading to identity confusion.

### 1.4. Identity and Criminal Behavior

Furthermore, identity confusion, resulting from disrupted identity development, can, in turn, give rise to criminal behavior. Theoretical frameworks suggest that identity problems heighten the risk of delinquency, particularly during adolescence ([26]; [46]; [60]). Empirical studies confirm this link ([2]; [17]; [45]). Moreover, identity confusion is tied to characteristics commonly associated with criminal behavior, such as impulsivity ([56]), low empathy and altruism ([57]), substance abuse ([2]), and emotional dysregulation ([1]). Identity confusion also correlates with personality traits like high neuroticism and low agreeableness ([22]), as well as personality disorder symptoms ([19]; [63]). Additionally, low self-control, a trait often associated with identity difficulties, strongly predicts criminal behavior ([65]) and is central to [31]’s ([31]) general theory of crime.

### 1.5. Can Identity Link Childhood Trauma to Criminal Behavior?

While childhood trauma is clearly associated with identity confusion, and identity confusion to criminal behavior, the potential mediating role of identity formation in this relationship remains underexplored. Prior research has investigated identity formation as a mediator in contexts such as psychopathology and criminal behavior ([12]) and childhood neglect and adult sexual disturbances ([7]; [53]). However, its mediating role in the specific link between childhood trauma and criminal behavior has not been thoroughly investigated. Understanding this mediation could provide valuable insights for intervention strategies targeting identity development to prevent criminal behavior in trauma-affected individuals.

### 1.6. The Present Study

The present study aims to investigate whether identity formation mediates the relationship between childhood trauma and criminal behavior. Identity formation is categorized into three variables: consolidated identity, disturbed identity, and lack of identity. Criminal behavior is operationalized as a grouping variable comprising a community sample and a sample of forensic psychiatric patients from high-security forensic psychiatric centers.

Based on the literature, four hypotheses are proposed. First, we hypothesize that greater childhood trauma will be more prevalent in the forensic psychiatric group compared to the community participants (H1). Second, we hypothesize that greater childhood trauma is expected to correlate with higher scores on disturbed identity and lack of identity and lower scores on consolidated identity (H2). Third, we hypothesize that disturbed identity and lack of identity will be more pronounced in the forensic psychiatric group, while consolidated identity will be higher in the community group (H3). Finally, building on evidence linking childhood trauma to identity problems and identity problems to criminal behavior, we hypothesize that all three identity variables—disturbed identity, lack of identity, and consolidated identity—will mediate the relationship between childhood trauma and criminal behavior (H4). To account for potential confounders, such as age differences and the heterogeneity of psychiatric diagnoses in the forensic subsample, we will control for age and overall personality dysfunction.

## 2. Materials and Methods

### 2.1. Participants

The initial study sample comprised 312 participants: 222 (53.6% females) community members and 90 male forensic psychiatric patients residing in high-security forensic psychiatric institutions. To ensure gender comparability, female community participants were excluded, along with two participants with over 30% missing data, resulting in a final sample of 191 males. This included 103 community participants (53.9%) and 88 forensic patients (46.1%), with ages ranging from 19 to 73 (*M* = 39.82 years, *SD* = 14.14). Nationalities included 64.4% Dutch, 30.4% Belgian, and 5.2% other nationalities, such as Turkish and Moroccan (for more details, see Table 1). Forensic patients had been institutionalized for an average of 127.64 months (range: 1–370; *SD* = 100.38) and committed various offenses, such as sexual offenses (45%), property offenses with or without violence (25%), violent crimes and possessions of weapons (21.6%), manslaughter (17%), murder (12.5%), and others like traffic or public disruption offenses (4.5%). Many patients committed multiple offenses. Diagnoses were diverse, including personality disorders (73.9%), substance use disorders (60.2%), paraphilic disorders (37.5%), psychotic disorders (25.0%), and others such as disruptive (10.2) or developmental disorders (9.1%), frequently with comorbidities.

### 2.2. Procedure

Patients from high-security forensic psychiatric centers in Gent, Antwerpen, and Rotterdam were invited to participate in this study. These centers treat individuals who have committed severe criminal offenses influenced by mental illness or personality disorders ([62]). Potential participants received an information letter from their unit head, emphasizing that their treatment would remain unaffected by their decision to participate. Information sessions were held to address questions, and participants were contacted two weeks later to provide informed consent. Consent included access to their personal data, such as clinical records, criminal history, and IQ. Participants were informed that their personal data would not be shared with others and that they had the right to withdraw from the study at any time without consequences. In addition to patient file information, self-report questionnaires were administered, with ample breaks provided to minimize fatigue. A research assistant was present on-site to assist with questions. Participants received a 10 EUR reward upon completion.

Community participants were recruited using snowball sampling and completed the same questionnaires online. Anonymous surveys were distributed by Master’s students. Inclusion criteria were being at least 18 years old, having adequate knowledge of the Dutch language, having no criminal convictions, and not undergoing psychological or psychotherapeutic treatment in the past three years. Informed consent was obtained, and participants retained the right to withdraw at any time. The ethical approval was obtained from the School of Social and Behavioral Sciences of Tilburg University and the Scientific Research Committee of FPC Kijvelanden (Nr EC-2017.45).

### 2.3. Measures

#### 2.3.1. Childhood Trauma

Childhood trauma was assessed using the Dutch version of the Childhood Trauma Questionnaire—Short Form (CTQ-SF; [6]), a retrospective self-report measure. The CTQ-SF consists of 28 items designed to assess five dimensions of childhood maltreatment: physical abuse (e.g., “Got hit so hard that I had to see a doctor or go to the hospital.”), emotional abuse (e.g., “People in my family said hurtful or insulting things to me.”), sexual abuse (e.g., “Someone tried to touch me in a sexual way/made me touch them.”), physical neglect (e.g., “I didn’t have enough to eat.”), and emotional neglect (“I felt loved [reverse-worded item].”). Each of these dimensions is measured with five items. Additionally, the CTQ-SF includes a minimization/denial scale with three items (e.g., “I had the perfect childhood [reverse-worded item]”), which was excluded from this study. All items were rated on a 5-point Likert scale ranging from never true (1) to very often true (5). A total score was calculated for the 25 clinical items, ranging from 25 to 125, with higher scores reflecting greater experiences of childhood trauma. The CTQ-SF demonstrated good internal consistency in prior research ([6]). In this study, its reliability was excellent (α = 0.96).

#### 2.3.2. Identity

Identity was measured using the Dutch version of the Self-Concept and Identity Measure (SCIM; [10]; [33]), a self-report questionnaire that evaluates three identity subscales: identity consolidation (10 items; e.g., “I know what I believe or value”), identity disturbance (11 items; e.g., “I imitate other people instead of being myself”), and lack of identity (six items; e.g., “I am broken”). Participants rated all 27 items on a 7-point Likert scale ranging from completely disagree (1) to completely agree (7). The mean scores of each subscale were calculated by summing the item scores and dividing by the number of items, with possible scores ranging from 1 to 7. Higher scores on each of the identity scales indicated more characteristics of that identity. [10] ([10]) reported acceptable to good internal consistency for the SCIM subscales across different samples: consolidated identity = 0.65 to 0.71, disturbed identity = 0.81 to 0.85, and lack of identity = 0.87 to 0.92. In the current study, reliability was questionable for consolidated identity (α = 0.60, mean inter-item correlation = 0.16) but good for disturbed identity (α = 0.85) and lack of identity (α = 0.83). As removing items did not improve the internal consistency of consolidated identity, the mean inter-item correlation was evaluated and deemed acceptable, ranging between 0.15 and 0.50 ([15]).

#### 2.3.3. Personality Dysfunction

Personality dysfunction was assessed using the Dutch version of the Personality Inventory for DSM-5—Short Form (PID-5-SF; [41]; [61]), a 100-item measure derived from the 220-item PID-5. The PID-5-SF evaluates dysfunctional personality traits across five domains: antagonism (24 items; “I’m better than almost everyone else.”), detachment (20 items; “I don’t get emotional.”), negative affectivity (24 items; “Plenty of people are out to get me.”), disinhibition (20 items; “I have no limits when it comes to doing dangerous things”), and psychoticism (12 items; “Things around me often feel unreal, or more real than usual.”). Participants rated items on a four-point Likert scale from 1 (very false or often false) to 4 (very true or often true). The average total score was calculated, ranging from 0 to 3, with higher scores indicating greater personality dysfunction. Previous research demonstrated good internal consistency (α = 0.84) ([4]), while the internal consistency in this study was excellent (α = 0.96).

#### 2.3.4. Criminal Behavior

Criminal behavior was operationalized as a grouping variable (0 = community participants, 1 = forensic participants). In this study, forensic patients were individuals who had committed serious offenses involving violence or threat of violence. These offenses included possession of arms, power by force, moral offenses with adults as victims, manslaughter, arson, and premeditated murder.

### 2.4. Statistical Analysis

All analyses were conducted using SPSS Statistics 29. Before analysis, the statistical assumptions of linearity, absence of outliers, multicollinearity, and independence of observations were checked. Linearity was tested using the Box and Tidwell procedure, which is appropriate for a dichotomous outcome variable. All predictors were linearly related to the logit of the dependent variable, meeting this assumption. Outliers were identified using z-scores, revealing three outliers on childhood trauma, two on lack of identity, one on consolidated identity, and one on personality dysfunction, totaling seven. These mild outliers were deemed natural variations and retained in the analysis. Multicollinearity was assessed using variance inflation factor (VIF) and tolerance values, with all predictors showing acceptable VIF values (VIF < 10) and tolerance (>0.1), confirming no violation of this assumption. The independence of the observations was ensured by the absence of overlap between groups. Our sample size was sufficient to address the research question, as determined by a power analysis using G*power version 3.1.9.7. The analysis indicated that a minimum sample size of 98 participants was required to detect a medium effect (*f*^2^ = 0.15), with a power of β = 0.80 and a significance criterion of α = 0.05. Our sample of 191 participants exceeded this requirement.

Descriptive statistics summarized the sample characteristics. Associations between the grouping variable and continuous variables were assessed using point biserial correlation, while Pearson’s correlation tested associations between continuous variables.

Lastly, a mediation analysis was conducted using PROCESS macro model 4 to test the study hypotheses. Childhood trauma was the independent variable; criminal behavior the dependent variable; and consolidated identity, disturbed identity, and lack of identity served as parallel mediators. First, the direct effects of childhood trauma on criminal behavior (H1), childhood trauma on each mediator (H2), and mediators on criminal behavior (H3) were tested. Subsequently, indirect effects of the mediators (H4) were tested using bootstrapping with 95% confidence intervals (CIs) to determine significance. Age and personality dysfunction were controlled for in all analyses.

## 3. Results

### 3.1. Descriptives

Table 2 provides an overview of the means and standard deviations for all study variables across the total sample and two subsamples. Correlations between the variables are shown in Table 3. Criminal behavior (versus no criminal behavior) was positively associated with childhood trauma, disturbed identity, lack of identity, and personality dysfunction. In addition, childhood trauma correlated positively with disturbed identity, lack of identity, and personality dysfunction. Consolidated identity negatively correlated with lack of identity and personality dysfunction but was positively associated with age. Lack of identity was positively associated with personality dysfunction, while personality dysfunction was negatively associated with age. Detailed subsample correlations between the variables can be found in Table A1 and Table A2 in Appendix A.

### 3.2. Results of Hypotheses Testing

All regression models in the mediation analysis, examining the association between childhood trauma and criminal behavior through the three identity subscales (consolidated, disturbed, and lack of), while controlling for age and personality dysfunction, were statistically significant (see Table 4).

The results showed that childhood trauma (B = 0.121, SE = 0.021, 95% CI [0.081, 1.161]) had a significant effect on criminal behavior (H1). Additionally, childhood trauma significantly influenced the lack of identity (B = 0.017, SE = 0.003, 95% CI [0.011, 0.023]) but showed no effect on the other two identity variables (H2). These findings suggest that individuals with higher childhood trauma scores are more likely to belong to the forensic group rather than the community group and exhibit higher levels of lack of identity. Furthermore, lack of identity (B = 0.971, SE = 0.332, 95% CI [0.321, 1.162]) significantly affected criminal behavior, while consolidated identity and disturbed identity did not (H3). This implies that individuals with a higher lack of identity are more likely to belong to the group of individuals convicted of a crime compared to the individuals from the general population.

Lastly, lack of identity was the only significant mediator in the association between childhood trauma and criminal behavior (B = 0.017, SE = 0.008, 95% CI [0.005, 0.035]) (H4). This suggests that individuals experiencing more childhood trauma are more likely to display a diminished sense of self, which subsequently increases the likelihood of belonging to the group of individuals who committed a crime. A summary of the unstandardized coefficients from the mediation model is provided in Table 5 and Figure 1.

## 4. Discussion

The association between childhood trauma and criminal behavior is well documented, but the role of identity formation remains less investigated in this relationship. The present study investigated whether three identity variables, including consolidated identity, disturbed identity, and lack of identity, mediate this relationship. The sample encompassed adult males from community and forensic psychiatric settings in Belgium and the Netherlands. Criminal behavior was assessed as a binary outcome, distinguishing forensic patients from community participants. Age and overall personality dysfunction were controlled to ensure a robust analysis.

First, the results of this study confirmed that childhood trauma is strongly associated with criminal behavior, supporting the first hypothesis and aligning with previous research ([8]; [35]; [44]). Additionally, as expected, individuals with a history of childhood trauma were more likely to exhibit a lack of identity, consistent with the literature suggesting that trauma disrupts identity development, leading to identity confusion and fragmentation ([5]; [53]). However, childhood trauma did not predict consolidated identity or disturbed identity, partially supporting the second hypothesis. Despite the non-significant effect in the mediation model, disturbed identity showed a positive bivariate association with childhood trauma and overall personality dysfunction. This lack of significance in the mediation model may be attributed to a shared variance between disturbed identity and other controlled factors, such as personality dysfunction. In contrast, consolidated identity did not show a significant bivariate association with childhood trauma, although the trend was in the expected direction. A plausible explanation is that the SCIM scale may not adequately capture the full complexity of consolidated identity, as evidenced by its questionable reliability in our sample. Previous research also found this subscale to be the least reliable among the SCIM measures ([10]). This low reliability could stem from differences in how consolidated identity manifests in clinical versus non-clinical populations ([33]). To address this limitation, future research should test for measurement invariance across these subsamples to ensure that the consolidated scale accurately measures the construct in both groups. Additionally, expanding measurement tools to include broader or alternative conceptualizations of identity, such as narrative identity, may better capture consolidated identity’s multidimensional nature. This would provide more robust insights into its relationship with childhood trauma and criminal behavior ([43]).

Furthermore, the results showed that only a lack of identity was significantly associated with criminal behavior, suggesting that individuals without a coherent sense of self are more likely to belong to the forensic psychiatric group than the community group. While disturbed identity showed a significant point biserial correlation with criminal behavior, this association became non-significant in the regression model. This outcome likely reflects the strong correlation between disturbed identity and lack of identity, as well as their shared variance with personality dysfunction, which was controlled for in the analysis. This supports the notion that, compared to a disturbed identity, a lack of identity is more unstable and undefined ([33]), exerting a greater influence on criminal behavior and choices. Consolidated identity showed no relationship with criminal behavior, further leading to the partial support of our third hypothesis. These findings align with previous research linking identity confusion to criminal behavior ([2]; [17]; [45]). Notably, the findings are consistent with [12] ([12]), who observed no direct effect of identity integration on criminal behavior but identified an indirect effect mediated by self-control. This difference suggests that the absence of self-control in our study may have influenced the findings. Alternatively, the low reliability of the SCIM’s consolidated identity subscale also likely contributed, as it may not fully capture the multidimensional complexity of the construct. For example, [13] ([13]) highlighted the significant role of emotional distress related to identity, such as anxiety, depression, and obsession, in influencing criminal behavior. [53] ([53]) also emphasized identity distress as an important factor that warrants further research.

A lack of identity emerged as the sole factor explaining the relationship between childhood trauma and criminal behavior, suggesting that childhood trauma may lead to feelings of emptiness and confusion about one’s identity, thereby increasing the likelihood of engaging in criminal behavior. Since consolidated and disturbed identity did not mediate this relationship, the fourth hypothesis was only partially supported. While limited research directly examines identity as a mediator in this relationship, the findings partially diverge from the existing literature, suggesting a broader role of identity in this context ([7]; [53]). In addition to the previously discussed limitations of our consolidated scale and shared variance among constructs, it could be that the lack of identity was the only mediator because it encompasses more than just identity-related struggles or reliance on others (i.e., disturbed identity). The lack of identity involves inner emptiness and a sense of non-existence, reflecting a more severe disruption in identity formation ([9], [11]) and, as such, a more severe psychopathology than identity disturbance. Thus, it is not surprising that lack of identity appeared as the only explanatory mechanism in the link between childhood trauma and criminal behavior, given that greater childhood trauma is linked to more severe psychopathology ([20]; [49]), which, in turn, is more strongly associated with criminal behavior ([58]). In brief, our findings highlight that severe psychopathology, like a lack of identity, may play a pivotal role in linking early adversity to later criminal behavior.

The findings of this study should be interpreted with caution due to several limitations. First, reliance on self-report questionnaires introduces potential bias, such as social desirability, response distortion, and malingering, particularly among forensic patients ([54]). Second, the consolidated identity subscale of the SCIM demonstrated low reliability, raising concerns about the accuracy of this measurement. Third, the study included only male participants, restricting the generalizability of the findings to females. Lastly, as a cross-sectional study, this research provides a snapshot in time and cannot establish causality. For example, while the findings suggest a link between childhood trauma, lack of identity, and criminal behavior, they cannot confirm whether childhood trauma causes identity disruption, which subsequently leads to criminal behavior. Despite this limitation, cross-sectional mediation models are valuable for identifying potential mechanisms and generating hypotheses for future research.

In conclusion, this study supports the literature that childhood trauma can disrupt healthy development and is closely linked to criminal behavior. Specifically, childhood trauma was associated with a lack of identity and criminal behavior, while no significant links were found for disturbed and consolidated identity. Lack of identity emerged as the only mediator in the relationship between childhood trauma and criminal behavior, highlighting the critical role of this identity dimension in understanding the link. Interventions designed to help individuals explore and define their identity may prove effective in reducing criminal behavior among those with trauma histories. Our findings suggest that addressing identity deficits may reduce identity disturbance, while focusing on a consolidated identity should not be prioritized, given its insignificant effects. Longitudinal research is needed to confirm these findings and establish causality.

## Figures and Tables

**Figure 1 behavsci-15-00056-f001:**
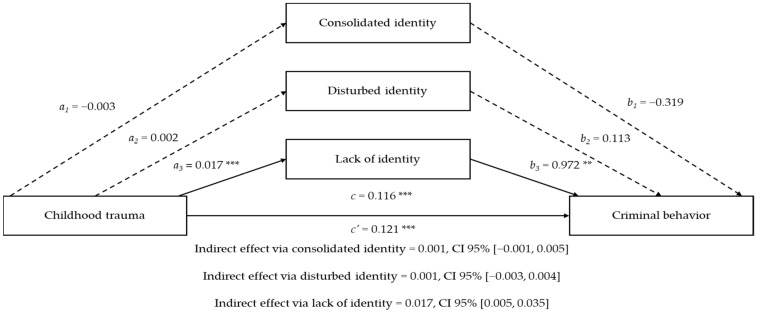
A three-variable mediation analysis with unstandardized regression coefficients. Criminal behavior was coded as a binary outcome, with 0 representing the community sample and 1 representing the forensic sample. The analysis was adjusted for age and dysfunctional personality. ** *p* < 0.001 and *** *p* < 0.0001.

**Table 1 behavsci-15-00056-t001:** Sample characteristics.

	Entire Sample (*N* = 191)	Forensic Sample (*n* = 88)	Community Sample (*n* = 103)
Variable	*M (SD)/N (%)*		
Age	39.82 (14.14)	41.82 (10.34)	38.11 (16.58)
Nationality			
Dutch	123 (64.4%)	27 (30.7%)	96 (93.2%)
Belgian	58 (30.4%)	58 (65.9%)	0 (0%)
Other	10 (5.2%)	3 (3.4%)	7 (6.8%)
Education			
Primary school	11 (5.8%)	11 (12.5%)	0 (0%)
Lower secondary/VMBO	18 (9.5%)	11 (12.5%)	7 (6.8%)
Higher secondary/HAVO	31 (16.2%)	27 (30.7%)	4 (3.9%)
VWO	6 (3.1%)	0 (0%)	6 (5.8%)
MBO	24 (12.6%)	0 (0%)	24 (23.3%)
HBO	37 (19.4%)	3 (3.4%)	34 (33%)
University	30 (15.7%)	2 (2.3%)	28 (27.2%)
Other	11 (5.8%)	11 (12.5%)	0 (0%)
Missing	23 (12%)	23 (26.1%)	0 (0%)

Note. *n* = number of participants; *M* = mean; *SD* = standard deviation.

**Table 2 behavsci-15-00056-t002:** Overview of the questionnaire characteristics.

	Entire Sample (*N* = 191)	Forensic Sample (*n* = 88)	Community Sample (*n* = 103)
Variable (Range)	*M (SD)*		
Childhood trauma (25–125)	47.00 (22.69)	62.38 (24.47)	34.00 (8.78)
Consolidated identity (1–7)	5.37 (0.81)	5.27 (0.81)	5.45 (0.80)
Disturbed identity (1–7)	2.97 (1.15)	3.25 (1.25)	2.74 (1.00)
Lack of identity (1–7)	2.63 (1.32)	3.18 (1.42)	2.16 (1.03)
Personality dysfunction (1–4)	1.89 (0.41)	1.96 (0.47)	1.83 (0.34)

**Table 3 behavsci-15-00056-t003:** Overview of the Pearson and point biserial correlations of the study variables (*N* = 191).

Variable	1.	2.	3.	4.	5.	6.	7.
1. Criminal behavior	—						
2. Childhood trauma	0.63 **	—					
3. Consolidated identity	−0.11	−0.10	—				
4. Disturbed identity	0.22 **	0.17 *	−0.14	—			
5. Lack of identity	0.38 **	0.42 **	−0.15 *	0.72 **	—		
6. Personality dysfunction	0.16 **	0.24 **	−0.20 **	0.67 **	0.68 **	—	
7. Age	0.13	0.06	0.26 **	−0.18 *	−0.04	−0.19 **	—

Note. * *p* < 0.05; ** *p* < 0.01.

**Table 4 behavsci-15-00056-t004:** Fit measures and overall model tests for regression models included in the mediation analysis.

Overall Model Test							
Model	Dependent Variable	R	R^2^	*F*	df1	df2	*p*
Mediator model							
	Consolidated identity	0.302	0.091	6.020	3	180	0.001
	Disturbed identity	0.659	0.434	46.090	3	180	<0.001
	Lack of identity	0.727	0.529	67.292	3	180	<0.001
**Full Model**		**−2LL**	**ModelLL**	**df**	** *p* **	**CoxSnell**	**Nagelkrk**
	Criminal behavior	135.482	117.833	6	0.000	0.473	0.633

Note. Models estimated the indirect, direct, and total effects of childhood trauma on criminal behavior, with consolidated identity, disturbed identity, and lack of identity as the mediating variables and age and personality dysfunction included as covariates.

**Table 5 behavsci-15-00056-t005:** Summary of the mediation analysis results.

Effect Type	Estimate	*SE*	95% Confidence Intervals		*p*
			Lower	Upper	
Indirect effect					
CT → Consolidated identity → CB	0.001	0.002	−0.001	0.005	
CT → Disturbed identity → CB	0.001	0.002	−0.003	0.004	
CT → Lack of identity → CB	0.017	0.008	0.005	0.035	
Path a					
CT → Consolidated identity	−0.003	0.003	−0.008	0.002	0.226
CT → Disturbed identity	0.002	0.003	−0.004	0.002	0.519
CT →Lack of identity	0.017	0.003	0.011	0.023	<0.001
Path b					
Consolidated identity → CB	−0.319	0.275	−0.859	0.220	0.246
Disturbed identity → CB	0.113	0.347	−0.568	0.794	0.744
Lack of identity → CB	0.972	0.332	0.321	1.622	0.003
Direct effect	0.121	0.021	0.081	0.161	<0.001
Covariates					
Age → Consolidated identity	0.013	0.004	0.004	0.021	0.003
Age → Disturbed identity	−0.007	0.005	−0.016	0.002	0.119
Age → Lack of identity	0.003	0.005	−0.006	0.013	0.498
Age → CB	0.023	0.016	−0.009	0.055	0.167
PD → Consolidated identity	−0.280	0.153	−0.581	0.023	0.070
PD → Disturbed identity	1.752	0.165	1.427	2.076	<0.001
PD → Lack of identity	1.979	0.176	1.632	2.326	<0.001
PD →CB	−2.732	0.927	−4.549	−0.916	0.003

Note. CT = childhood trauma; CB = criminal behavior (0 = community sample and 1 = forensic sample); PD = personality dysfunction.

## Data Availability

Data available on reasonable request.

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
