# Peer review of "Unraveling the Association: How Identity Mediates the Impact of Childhood Trauma on Criminal Behavior"

_behavsci, 2025, doi:10.3390/bs15010056_

Round 1

Reviewer 1 Report

Comments and Suggestions for Authors

This study successfully collected valuable data from a hard-to-reach population. I extend my gratitude to the research team for their efforts. I hope this study will be widely utilized as evidence for understanding childhood trauma, and I encourage the authors to consider the following suggestions to improve the quality of the manuscript.

1.

In this study, the variable "identity" is analyzed by categorizing it into three sub-factors: identity consolidation, identity disturbance, and lack of identity. However, there is insufficient explanation regarding how these three variables differ and what distinctions exist among them. If the three sub-factors are to be analyzed separately, as presented in the current results, it is essential to supplement the introduction with detailed explanations of these distinctions. It is challenging for readers to grasp the differences between the sub-factors. Additionally, the conclusion and discussion sections should be expanded to provide specific recommendations for each of the three sub-factors.

2. 

To enhance the readers' intuitive understanding, I recommend including a visual representation of the research model. Using solid lines to indicate significant effects and dotted lines for non-significant effects in the diagram would allow readers to comprehend the findings quickly and easily.

3.

303: Table 5
To improve the clarity of the table, it is necessary to indicate the significance levels (*) for the estimates. Currently, the lack of significance markers makes it difficult to determine the impact and significance of the estimates. It would also be helpful to include significance levels for the mediation effects. The current table requires readers to manually examine the confidence intervals, which is less efficient.
Example: * p < .05, ** p < .01, *** p < .001

4.

304: Note Error
CB = CT = Childhood trauma; Criminal behavior

Author Response

Comments and Suggestions for Authors

This study successfully collected valuable data from a hard-to-reach population. I extend my gratitude to the research team for their efforts. I hope this study will be widely utilized as evidence for understanding childhood trauma, and I encourage the authors to consider the following suggestions to improve the quality of the manuscript.

1.

In this study, the variable "identity" is analyzed by categorizing it into three sub-factors: identity consolidation, identity disturbance, and lack of identity. However, there is insufficient explanation regarding how these three variables differ and what distinctions exist among them. If the three sub-factors are to be analyzed separately, as presented in the current results, it is essential to supplement the introduction with detailed explanations of these distinctions. It is challenging for readers to grasp the differences between the sub-factors. Additionally, the conclusion and discussion sections should be expanded to provide specific recommendations for each of the three sub-factors.

Response: Thank you for your comment. We agree with your point. In response, we have clarified the definitions of each identity dimension and emphasized their differences. Please see lines 77-85.

Additionally, following your suggestion, we have included recommendations for each of the three identity dimensions in the conclusion and discussion section. See lines 364-366 and 417-421.

  1.  

To enhance the readers' intuitive understanding, I recommend including a visual representation of the research model. Using solid lines to indicate significant effects and dotted lines for non-significant effects in the diagram would allow readers to comprehend the findings quickly and easily.

Response: Thank you for your suggestion. As requested, we have included a visual representation of the research model. See line 318-322.

3.

303: Table 5
To improve the clarity of the table, it is necessary to indicate the significance levels (*) for the estimates. Currently, the lack of significance markers makes it difficult to determine the impact and significance of the estimates. It would also be helpful to include significance levels for the mediation effects. The current table requires readers to manually examine the confidence intervals, which is less efficient.
Example: * p < .05, ** p < .01, *** p < .001

Response: Thank you for your comment. We have added a column with p-values to Table 5 for all effects, except for the indirect effect, as its significance was determined using 95% confidence intervals.

4.

304: Note Error
CB = CT = Childhood trauma; Criminal behavior

Response: Thank you for bringing this to our attention. Indeed, this was an error, which we have corrected accordingly. See line 310.

Reviewer 2 Report

Comments and Suggestions for Authors

The paper seems to be adequately developed, with alignment between the purpose/aims, research questions, and results. 

A question could assist readers in understanding, though. Take for example, lines 286-303 list betas, but what is the magnitude and practical application of CT--> lack of identity or lack of identity as only significant mediator? As stated on lines 291-293, lack of identity is strong by itself.

Author Response

Comments and Suggestions for Authors

The paper seems to be adequately developed, with alignment between the purpose/aims, research questions, and results. 

A question could assist readers in understanding, though. Take for example, lines 286-303 list betas, but what is the magnitude and practical application of CT--> lack of identity or lack of identity as only significant mediator? As stated on lines 291-293, lack of identity is strong by itself.

Response: Thank you for your comment. First, to provide a clearer understanding of the three identity dimensions and their distinctions, we expanded their definitions in the introduction (see lines 77-85). Additionally, in the discussion section, we also emphasized the practical implications of addressing identity deficits, especially for individuals impacted by childhood trauma. Focusing on this issue can help reduce criminal behavior and likely alleviate identity disturbances. The latter is supported by the strong correlation observed between a lack of identity and disturbed identity, as well as the non-significant effects of disturbed identity when a lack of identity is present. Furthermore, we emphasized that a lack of identity represents a more severe impairment in self-concept than an identity disturbance, and as such, it has a stronger influence on criminal behavior. See lines 364-366 and 417-420.
